# Preparation and Modification of PVDF Membrane and Study on Its Anti-Fouling and Anti-Wetting Properties

Yubo Wang [1], Qiang Guo [2], Zhen Li [1], Jingfeng Li [2], Ruimin He [3], Kaiyang Xue [1] and Shuqin Liu [1,*]

1. School of Chemical & Environmental Engineering, China University of Mining & Technology, Beijing 100083, China; w15300220424@163.com (Y.W.); 18813095516@163.com (Z.L.); bambooleprechaun@163.com (K.X.)
2. State Key Laboratory of Water Resource Protection and Utilization in Coal Mining, China Energy Investment Corporation, Beijing 100011, China; guoqiang_2004@163.com (Q.G.); liuhuan@student.cumtb.edu.cn (J.L.)
3. China Energy Shenhua Shendong Coal Group Co., Ltd., Beijing 100045, China; hml13146888733@163.com
* Correspondence: liushuqin@cumtb.edu.cn

**Abstract:** Membrane distillation (MD) has unique advantages in the treatment of high-salt wastewater because it can make full use of low-grade heat sources. The high salinity mine water in western mining areas of China is rich in $Ca^{2+}$, $Mg^{2+}$, $SO_4^{2-}$ and $HCO_3^{-}$. In the MD process, the inorganic substances in the feed will cause membrane fouling. At the same time, low surface tension organic substances which could be introduced in the mining process will cause irreversible membrane wetting. To improve the anti-fouling and anti-wetting properties of the membrane, the PVDF omniphobic membrane in this paper was prepared by electrospinning. The water contact angle (WCA) can reach 153°. Direct contact membrane distillation (DCMD) was then used for treating high-salinity mine water. The results show that, compared with the unmodified membranes, the flux reduction rate of the omniphobic membrane was reduced by 34% in 20 h, showing good anti-fouling property. More importantly, the omniphobic membrane cannot be wetted easily by the feed containing 0.3 mmol/L SDS. The extended Derjaguin–Landau–Verwey–Overbeek (XDLVO) theory was used to analyze the free energy of the interface interaction between the membrane and pollutants, aiming to show that the omniphobic membrane was more difficult to pollute. The result was consistent with the flux variation in the DCMD process, providing an effective basis for explaining the mechanism of membrane fouling and membrane wetting.

**Keywords:** high salinity mine water; omniphobic membrane; electrospinning; XDLVO theory; membrane distillation

## 1. Introduction

For a long time, coal has occupied a dominant position in the energy production and consumption structure of China. As coal is deeply buried underground, its mining process will lead to the destruction of the aquifer. The discharge of mine water degrades the water quality and thus water cannot be directly used for potable water and some industrial applications [1–3]. Therefore, the treatment and resource utilization of mine water are particularly critical. The salt content of mine water discharged to the earth's surface should be less than 1000 mg/L, while high salinity mine water (TDS ≥ 1000 mg/L) widely exists in mining areas in Western China, and the water contains a large amount of $Ca^{2+}$, $Mg^{2+}$, $SO_4^{2-}$, $HCO_3^{-}$. Direct discharge of the mine water will lead to environmental problems such as land salinization and soil erosion. To realize the deep treatment and desalination of high salinity mine water, electrodialysis, reverse osmosis, distillation and ion exchange came into being. For example, ammonium and nitrate in mine water can be enriched 3.6 and 5.7 times by reverse osmosis technology, respectively [4]. Although the high salinity mine water treatment process, with reverse osmosis technology as the core,

has been relatively mature, problems such as high treatment cost and complexity still need to be solved.

　　MD is a membrane separation technology driven by the difference in the chemical potential of volatile components on both sides of the membrane. Due to the advantage of a theoretical salt rejection rate of 100%, combined with the low-grade heat source, MD has been used increasingly in the treatment of high salinity mine water in recent years [5–7]. Vacuum membrane distillation (VMD) is one of the most effective technologies of MD. In 170 h continuous operation by VMD, the salt rejection rate can reach 99.5%. However, membrane fouling and membrane wetting restrict the further application of MD in real life. Membrane fouling refers to the phenomenon that pollutants in the feed adhere to the membrane surface and membrane pores through physical deposition, electrostatic adsorption or chemical action in the process of the membrane distillation. According to the different causes, membrane fouling can be divided into inorganic fouling, organic fouling and biological fouling. Membrane wetting refers to the phenomenon that the pores of the hydrophobic membrane lose the ability to block the feed from passing through the membrane, and the feed can pass through the pore of the feed side to the infiltration side, leading eventually to feed pollution. It is a dynamic process in which the gas-liquid interface gradually moves towards the infiltration side. When treating high salinity mine water, the hydrophobic membrane used in MD is easily fouled by inorganic pollutants such as calcium carbonate and calcium sulfate [8], resulting in a significant decrease in flux. If organic pollutants with low surface tension (such as surfactants) permeate through the water, they will cause irreversible wetting to the membrane and reduce the salt rejection capability [9,10].

　　To prevent the membrane from fouling and wetting, methods such as pre-treatment (hardness removal) and post-treatment (membrane cleaning) can be adopted. However, the addition of $NaCO_3$, $NaHCO_3$ and other chemicals not only introduces new ions, but the hardness removal efficiency also cannot reach 100%. Using dilute hydrochloric acid, citric acid and other acidic substances to clean the polluted membrane also aggravates the risk of membrane wetting. Therefore, the goal of reducing membrane fouling can be achieved by optimizing the membrane itself without changing the operating conditions. The omniphobic PVDF membrane was developed by spraying a solution containing nano silica ($SiO_2$NPs) FAS, and FS (Zonyl@8867L) onto the surface of the PVDF membrane, achieving good anti-wetting and anti-fouling characteristics [11]. Researchers took inspiration from lotus leaves and realized the purpose of anti-fouling and anti-wetting by constructing a micro-nano-composite structure and introducing low surface energy substances onto the surface of the hydrophobic membrane.

　　In this paper, the PVDF omniphobic membrane was prepared by electrospinning. The addition of the low surface energy material gives the membrane the property of anti-fouling of inorganic pollutants and anti-wetting of surfactant in MD. The modified membrane was then used to treat high salinity mine water by DCMD. The XDLVO theory was used to analyze the interaction mechanism between calcium–magnesium salt, SDS and the membrane surface in the process of membrane fouling and wetting.

## 2. Materials and Methods

### 2.1. Preparation of PVDF-H Omniphobic Membrane

　　A commercial membrane (PVDF-H) was purchased from Millipore Sigma. The main parameters are shown in Table 1.

### 2.2. Preparation of PVDF Omniphobic Membrane

　　To begin, 12 wt% PVDF powder was added to 20 mL of solvent (DMAC/acetone = 4/1) to prepare solution 1. The solution was stirred vigorously at 60 °C for 12 h and was allowed to stand for 12 h for defoaming.

　　Afterwards, 0.1383 g $NH_3 \cdot H_2O$ and 0.1383 g (3, 3, 4, 4, 5, 5, 6, 6, 7, 7, 8, 8, 9, 9, 10, 10, 10-Heptadecafluorodecyl)trimethoxysilane (17-FAS) were added to 20 mL of solvent

(DMAC/acetone = 4/1), stirring for 30 min; 12 wt% PVDF powder was then added to prepare solution 2, The solution was stirred vigorously at 60 °C for 12 h and was allowed to stand for 12 h for defoaming.

**Table 1.** Parameters of PVDF-H membrane.

| Parameters | Values |
|---|---|
| Average pore size (μm) | 0.450 |
| Porosity (%) | 82 |
| Thickness (mm) | 0.150 |
| Contact angle (°) | 119.80 |
| Effective membrane surface area (m$^2$) | 0.0032 |

PVDF fiber membrane was prepared by electrospinning. The spinning voltage was +15 kV, −1 kV, the spinning speed was 0.8 mL/h, the receiving distance was 15 cm, the spinning temperature was 23 ± 1 °C, and the spinning humidity was 45 ± 2%. Firstly, solution 1 was spun for 7 h to prepare the unmodified membrane (PVDF-E), then solution 2 was spun on its surface for 4 h, and the omniphobic PVDF membrane (PVDF-O) was prepared and dried in the oven at 60 °C for 2 h to remove the residual solvent on the surface (Figure 1).

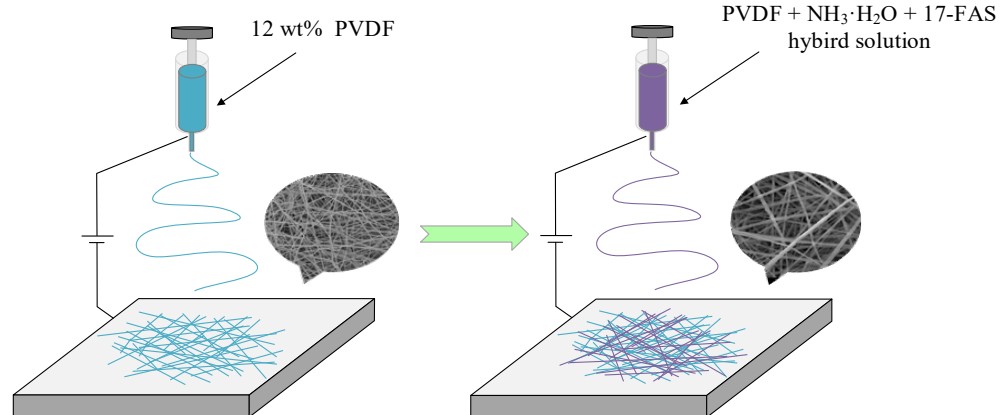

**Figure 1.** Fabrication procedures of the omniphobic membrane.

*2.3. Characterization and Theoretical Calculation of Membrane*

2.3.1. Characterization Method

The electrospinning equipment was purchased from Beijing Yongkang Leye Technology Development Co., Ltd. The equipment performs traditional solution electrospinning and the model is ET-2535X. The DCMD device (Figure 2) is self-assembled. The temperature of the feed is 60 °C and the temperature of the condensate is 15 °C. The device operates continuously with circulating feeding at a flow rate of 0.5 L/min.

The membrane flux J (kg/(m$^2$·h)) was calculated as

$$J = \frac{W}{S \times t} \tag{1}$$

where W is the mass of permeate water (kg); S is the membrane area (m$^2$); and t is the operation time (h).

The salt rejection rate R (%) was determined as

$$R = \left( \frac{\delta_0 - \delta_t}{\delta_0} \right) \times 100\% \tag{2}$$

where $\delta_0$ is the initial conductivity (μS/cm) of the feed, and $\delta_t$ is the conductivity (μS/cm) of the permeate water at time $t$.

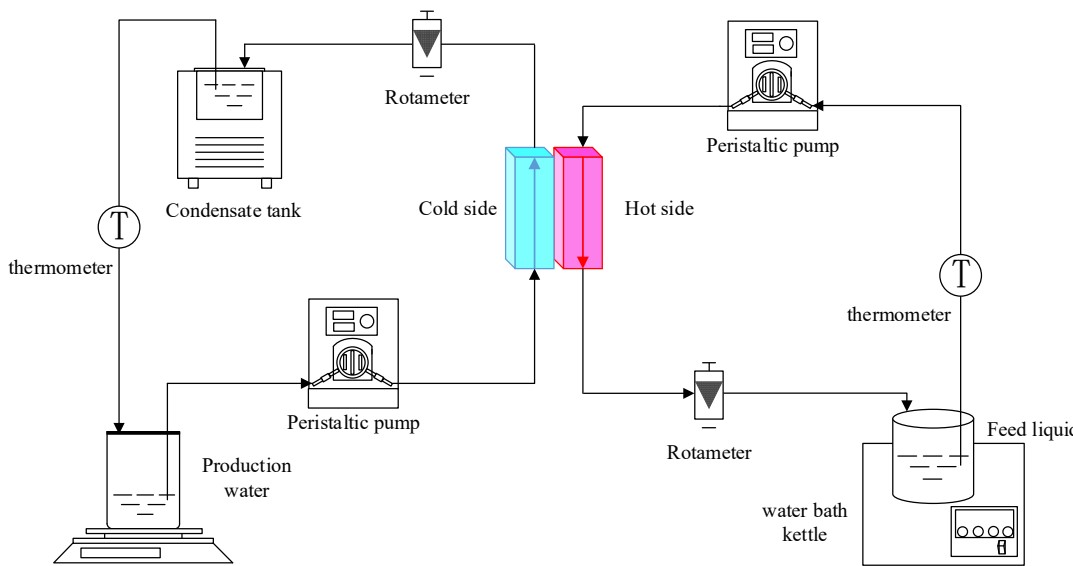

**Figure 2.** Schematic of the experimental DCMD system.

In order to explore the influence of the types and contents of pollutants in the feed on the experiment, we prepared the simulated water by adding different substances to the deionized water. The water consumption for each experiment is two liters, while the amount of components contained in each liter of deionized water and the parameters of the water are shown in Tables 2 and 3.

**Table 2.** Experimental water and water quality parameters.

| Hot Feed Composition (Per Liter of Deionized Water) | Parameters | |
|---|---|---|
| | Conductivity | pH |
| 4 g NaCl | $7.21 \times 10^3$ μs/cm | 7.19 |
| 35 g NaCl | $32.5 \times 10^3$ μs/cm | 6.93 |
| 0.0944 g CaCl$_2$ + 0.2164 g MgSO$_4$ + 0.1114 g NaHCO$_3$ | 581 μs/cm | 7.37 |
| 0.4718 g CaCl$_2$ + 1.082 g MgSO$_4$ + 0.557 g NaHCO$_3$ | $2.47 \times 10^3$ μs/cm | 7.86 |
| Mine water | $16.05 \times 10^3$ μs/cm | 8.12 |
| Concentrated mine water | $33.5 \times 10^3$ μs/cm | 7.88 |
| Mine water + 0.3 mmol SDS | $16.21 \times 10^3$ μs/cm | 8.12 |

**Table 3.** The main composition of the mine water and reverse osmosis mine water.

| Major Ion Composition | Ion Concentration (mg/L) | |
|---|---|---|
| | Mine Water | Concentrated Mine Water |
| Cl$^-$ | 1888.75 | 2567.25 |
| SO$_4^{2-}$ | 1880.23 | 3732.14 |
| Na$^+$ | 1957.63 | 2939.72 |
| HCO$_3^-$ | 563.25 | 713.27 |
| Ca$^{2+}$ | 313.65 | 445.65 |
| Mg$^{2+}$ | 356.82 | 496.82 |
| Al | 0.039 | 0.052 |
| NO$_3^-$ | 11.56 | 11.22 |
| Mn | 0.012 | 0.02 |
| CODCr | 21 | 27 |

### 2.3.2. Calculation of Interfacial Free Energy by XDLVO Theory

XDLVO theory was first proposed by van Oss to describe the interfacial free energy between two solid surfaces quantitatively [12]. According to this theory [13], the total interfacial free energy ( $\Delta G^{TOT}$ ) is the sum of van der Waals free energy ( $\Delta G^{LW}$ ), Lewis acid-base free energy ( $\Delta G^{AB}$ ) and free energy of an electrostatic double layer ( $\Delta G^{EL}$ ). The mathematical relation is shown as Equations (3)–(6).

$$\Delta G^{TOT} = \Delta G^{LW} + \Delta G^{AB} + \Delta G^{EL} \tag{3}$$

$$\Delta G^{LW} = \Delta G_{d0}^{LW} \frac{d_0^2}{h^2} \tag{4}$$

$$\Delta G^{AB} = \Delta G_{d0}^{AB} \exp\left[\frac{d_0 - h}{\lambda}\right] \tag{5}$$

$$\Delta G^{EL}(h) = \varepsilon_0 \varepsilon_r \kappa \xi_f \xi_m \left[\frac{\xi_f^2 + \xi_m^2}{2\xi_f \xi_m}(1 - coth\kappa h) + \csc h(\kappa h)\right] \tag{6}$$

where the subscripts m, w and f represent flat membrane, aqueous solution and pollutants respectively. $d_0$ is the minimum action distance between the pollutant and the membrane surface; $\lambda$ is the attenuation length of polar force AB in aqueous solution, $\lambda$ = 0.6 nm; $\xi$ is the zeta potential of pollutants and the membrane surface; $\varepsilon_0$ and $\varepsilon_r$ is the vacuum dielectric constant ($8.854 \times 10^{-12}$ C$^2$/J·m) and the relative dielectric constant of aqueous solution (78.4); $\Delta G_{d0}^{LW}$ , $\Delta G_{d0}^{AB}$ and $\Delta G_{d0}^{EL}$ respectively represent the interface free energy per unit area ($d_0$ = 0.158 nm) when the two solid planes are in contact with the minimum separation distance. The calculation formulas are as follows (Equations (7)–(9)).

$$\Delta G_{do}{}^{LW} = -2(\sqrt{\gamma_m{}^{LW}} - \sqrt{\gamma_w{}^{LW}})(\sqrt{\gamma_f{}^{LW}} - \sqrt{\gamma_w{}^{LW}}) \tag{7}$$

$$\Delta G_{do}{}^{AB} = 2[\sqrt{\gamma_w{}^+}(\sqrt{\gamma_f{}^-} + \sqrt{\gamma_m{}^-} - \sqrt{\gamma_w{}^-}) + \sqrt{\gamma_w{}^-}(\sqrt{\gamma_f{}^+} + \sqrt{\gamma_m{}^+} - \sqrt{\gamma_w{}^+}) - (\sqrt{\gamma_f{}^-\gamma_m{}^+} + \sqrt{\gamma_f{}^+\gamma_m{}^-})] \tag{8}$$

$$\Delta G_{d0}{}^{EL} = \varepsilon_0 \varepsilon_r \kappa \xi_f \xi_m \left[\frac{\xi_f^2 + \xi_m^2}{2\xi_f \xi_m}(1 - coth\kappa h) + \csc h(\kappa h)\right] \tag{9}$$

where $\gamma^{LW}$ , $\gamma^-$ , $\gamma^+$ represent the sub-item of van der Waals, electron donor and electron acceptor surface tension, mJ·m$^{-2}$; $\gamma_m^{LW}$ , $\gamma_m^+$ , $\gamma_m^-$ respectively represents the sub-item of membrane surface tension, mJ·m$^{-2}$, $\gamma_f^{LW}$ , $\gamma_f^+$ , $\gamma_f^-$ respectively represents the sub-item of pollutant surface tension, mJ·m$^{-2}$; $\kappa$ represents the reciprocal of Debye constant, nm$^{-1}$. The calculation formulas are as follows (Equations (10)–(13)):

$$\kappa = \sqrt{\frac{e^2 \sum n_i Z_i^2}{\varepsilon_0 \varepsilon_r kT}} \tag{10}$$

$$\frac{(1 + \cos\theta_0)}{2}\gamma_l{}^{TOT} = \sqrt{\gamma_l{}^{LW}\gamma_s{}^{LW}} + \sqrt{\gamma_l{}^-\gamma_s{}^+} + \sqrt{\gamma_l{}^+\gamma_s{}^-} \tag{11}$$

$$\gamma^{AB} = 2\sqrt{\gamma^+\gamma^-} \tag{12}$$

$$\gamma^{TOT} = \gamma^{LW} + \gamma^{AB} \tag{13}$$

where $e$ is the amount of electron charge ($1.6 \times 10^{-19}$ C), $Z_i$ represents the valence of ion i, $n_i$ represents the molar concentration of ion i in the solution, and k represents the Boltzmann constant ($1.38 \times 10^{-23}$ J/K); T is the absolute temperature (K). $\theta_0$ represents the intrinsic contact angle of different test liquids on the surface of the test sample, and the subscripts s

and l represent the solid surface and the test liquid respectively. In this paper, deionized water, glycerol and di-iodomethane are used as test liquids. The surface tension parameters of the three test liquids are listed in Table 4.

**Table 4.** Surface tension parameters of the three test liquids.

| Liquid | $\gamma^{LW}$ | $\gamma^+$ | $\gamma^-$ | $\gamma^{TOT}$ |
|---|---|---|---|---|
| deionized water | 21.8 | 25.5 | 25.5 | 72.8 |
| glycerine | 34.0 | 3.9 | 57.4 | 64.0 |
| diiodomethane | 50.8 | 0.0 | 0.0 | 50.8 |

When measuring the contact angles, the surface roughness of the membrane will have a certain impact on the measurement results. Therefore, when using Equation (11) to calculate the surface tension, the measurement results of the contact angles need to be corrected. The formula is as follows (equation (14)):

$$(1 + \frac{\cos\theta_0}{1+SAD})\gamma_l^{TOT} = 2(\sqrt{\gamma_l^{LW}\gamma_s^{LW}} + \sqrt{\gamma_l^-\gamma_s^+} + \sqrt{\gamma_l^+\gamma_s^-}) \tag{14}$$

where *SAD* is the deviation value of membrane surface area, which can be measured by AFM.

## 3. Results and Discussion

### 3.1. Anti-Fouling and Anti-Wetting Performance Test of Omniphobic Membrane

3.1.1. Morphology, Composition and Properties of Omniphobic Membrane

The SEM images of the PVDF-H membrane, PVDF-E membrane and PVDF-O membrane are shown in Figure 3. The fibers of the PVDF-O membrane are stacked in a disorderly fashion, and the introduction of inert ammonia solvent leads to coarser membrane fibers. Therefore, the roughness of the PVDF-O membrane is much higher than that of the PVDF-H and PVDF-E membranes. Si element was detected on the surface of the PVDF-O membrane by EDS, and the F/C ratio was greatly improved, which proved that PVDF and 17-FAS were crosslinked (Table 5).

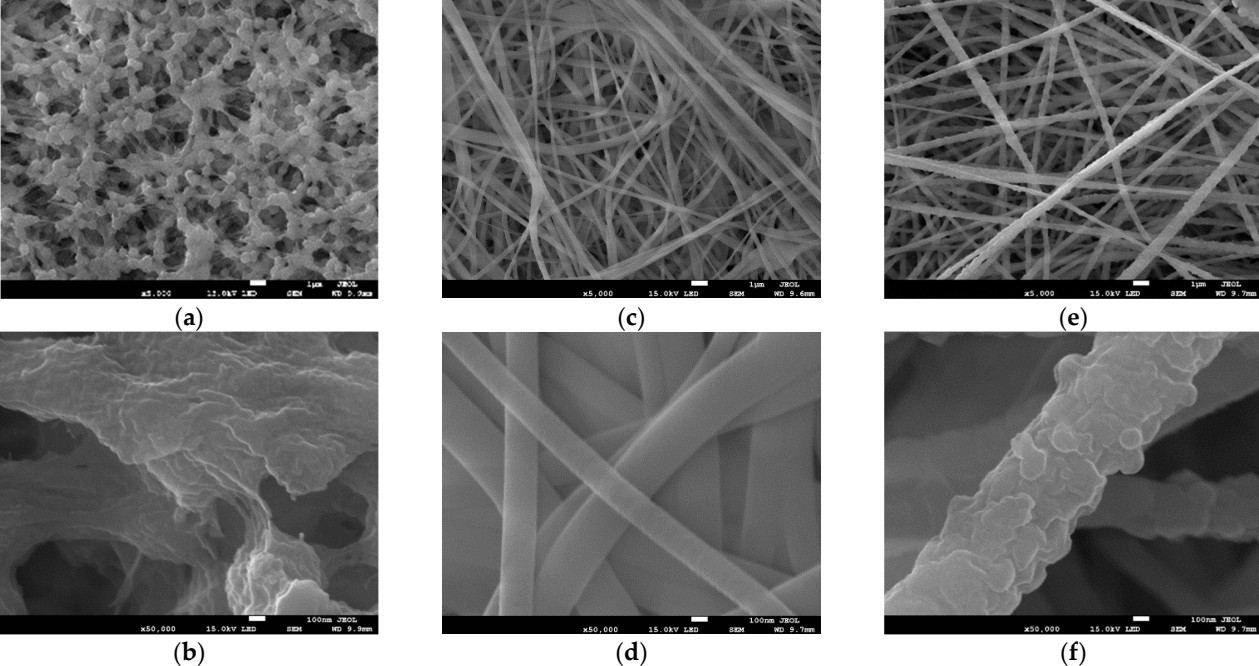

**Figure 3.** Scanning electron micrograph of (**a**) and (**b**) PVDF-H, (**c**) and (**d**) PVDF-E, (**e**) and (**f**) PVDF-O membrane.

**Table 5.** The content of each element on the surface of PVDF-H, PVDF-E and PVDF-O membranes tested by EDS.

| Element | The Content of Each Element on the Surface of the Membranes | | |
|---|---|---|---|
| | **PVDF-H** | **PVDF-E** | **PVDF-O** |
| C | 66.7% | 64.9% | 52.94% |
| F | 29.1% | 28.7% | 40.73% |
| Na | 0 | 0 | 0 |
| Cl | 0 | 0 | 0 |
| Ca | 0 | 0 | 0 |
| Mg | 0 | 0 | 0 |
| O | 0 | 0 | 0.08% |
| S | 0 | 0 | 0 |
| Si | 0 | 0 | 0.22% |

The changes of functional groups on the membrane surface were characterized by ATR-FTIR and the spectra are presented in Figure 4. As for the PVDF-H fibrous membrane, there is an obvious characteristic absorption peak at 876 cm$^{-1}$, which represents the skeleton vibration of C-C. A strong absorption peak of C-H appeared at 1405 cm$^{-1}$, and there are two characteristic absorption peaks at 838 cm$^{-1}$ and 1071 cm$^{-1}$, which can be attributed to crystallization induction $\alpha$ of PVDF. For the PVDF-O membrane, two absorption peaks appeared near 1172 cm$^{-1}$ and 1233 cm$^{-1}$, which were assigned to the symmetric stretching and anti-symmetric stretching vibration of C-F respectively, and the bending vibration absorption peak of C-F was found near 762 cm$^{-1}$. The intensity of the absorption peak of C-F is higher than that of the PVDF-H membrane, indicating that the proportion of F element on the membrane surface was enhanced and the omniphobic modification of the membrane was realized.

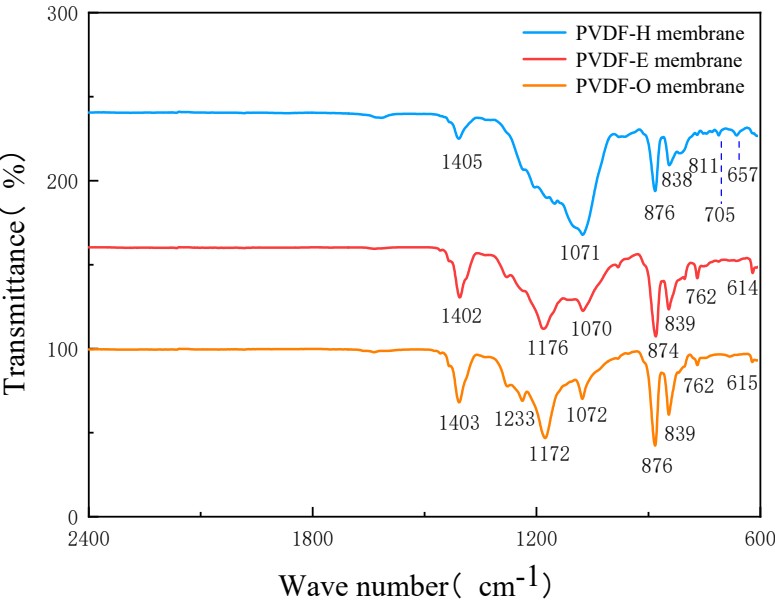

**Figure 4.** ATR-FTIR spectra of the PVDF-H, PVDF-E and PVDF-O membrane.

In general, the membrane surface CA can be utilized to evaluate the wettability of the membrane surface. As shown in Figure 5, the contact angles of the PVDF-H and PVDF-O membranes with different liquids in air medium are compared. The results show that the WCA of the PVDF-O membrane is 153°, realizing the superhydrophobic effect. No matter what kind of test liquid is used, the contact angle of the PVDF-O membrane is higher than that of the PVDF-H membrane. Attributing this to the low surface energy property of

PVDF-O membrane, hexadecane and absolute ethanol droplets cannot wet the membrane immediately when they contact the surface.

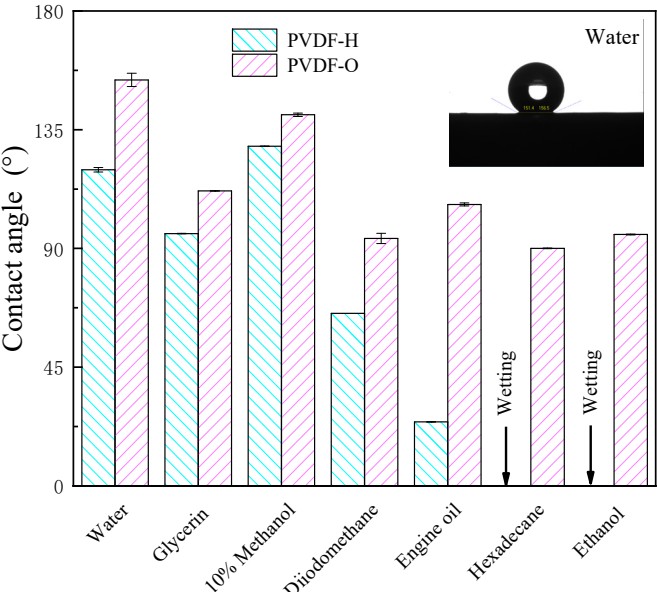

**Figure 5.** Contact angles of different liquids on the surface of PVDF-H and PVDF-O membrane.

In the process of preparing omniphobic membranes by electrospinning, the addition of ammonia to the spinning solution plays four key roles [14,15]: (1) As shown in Figure 3f, the addition of inert ammonia solvent makes the membrane fiber rough and also improves the roughness of the membrane surface, making the hydrophobic property of the membrane better; (2) The addition of ammonia encourages local microphase separation, causing crosslinking and clustering in the PVDF chain, which improves the mechanical strength of the polymer membrane in the spinning process; (3) Ammonia can order the hydroxyl functionalization of PVDF under the action of high-temperature etching. The introduction of hydroxyl provides active sites for PVDF, which is conducive to its cross-linking with 17-FAS; (4) The hydrolysis effect of 17-FAS in the aqueous solution is poor. Ammonia can provide an alkaline environment and accelerate the hydrolysis process of 17-FAS.

### 3.1.2. Anti-Fouling Process of Omniphobic Membrane

The fouling process of inorganic salt is shown in Figure 6. The mechanism involves the formation and growth of a crystal nucleus. From the perspective of dynamics, calcium and magnesium ions in mine water can be removed as much as possible through pre-treatment or other methods to reduce the hardness of water, which can prolong the cycle from unsaturated solution to supersaturated solution, and then delay the formation time of a crystal nucleus on the membrane surface, reducing the effect of fouling on flux and prolonging the service life of the membrane. From a thermodynamic point of view, improving the hydrophobic properties of the membrane surface can increase the Gibbs free energy of heterogeneous nucleation on the membrane surface. More work is needed in order to form a nucleus on the membrane surface.

All three kinds of membranes were used to treat high salinity mine water by DCMD, as shown in Figure 7. The main inorganic pollutants in high salinity mine water include alkaline scaling ($CaCO_3$), non-alkaline scaling ($CaSO_4$) and a small amount of magnesium salt [16]. The formation of $CaCO_3$ is affected by factors such as pH and feed liquid temperature. By contrast, the chemical properties of $CaSO_4$ are more stable, which may result in more serious pollution to the membrane [17]. Attributing this to the existence of the hydrophobic layer, the thickness of the PVDF-O membrane is higher than that of the PVDF-E membrane, which increases the mass and heat transfer resistance of water vapor. Therefore, the flux of the PVDF-O membrane is slightly lower than that of the

PVDF-E membrane. After 20 h operation of DCMD, the flux of the PVDF-H and PVDF-E membranes decreased by 62.5% and 38.3% respectively, while the flux of the PVDF-O membrane decreased by only 28.5%, and the salt rejection rate was as high as 99.9%, indicating that omniphobic membrane reduced the fouling of the membrane surface to a certain extent. However, due to the limitation of mechanical strength, the properties of the PVDF-E membrane prepared by electrospinning decreased significantly, and the salt rejection rate was also reduced apparently.

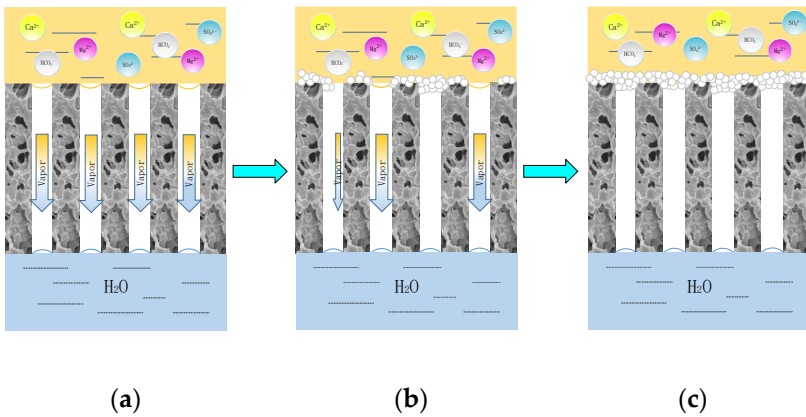

(**a**)   (**b**)   (**c**)

**Figure 6.** Inorganic fouling mechanism of the membrane: (**a**) Not fouled, (**b**) Partially fouled, (**c**) Completely fouled.

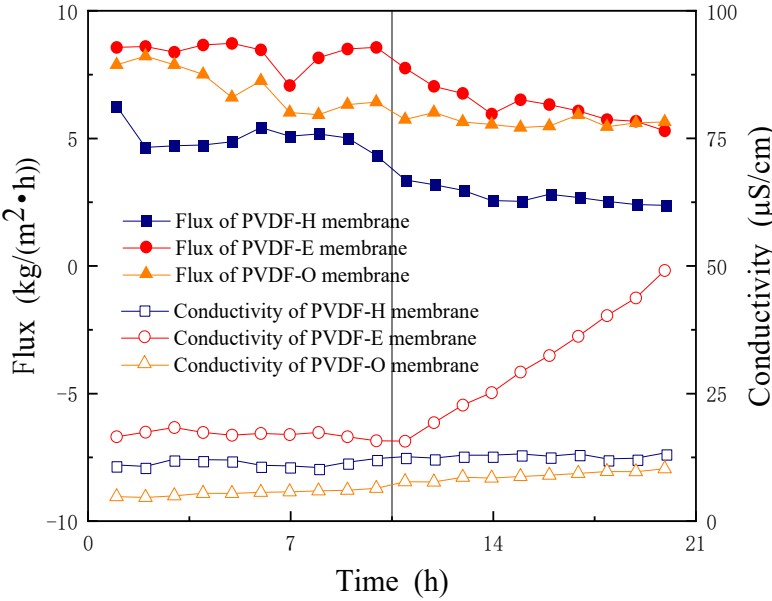

**Figure 7.** Flux and conductivity variation of PVDF-H, PVDF-E and PVDF-O membranes in the treatment of high-salinity mine water by DCMD.

### 3.1.3. Anti-Wetting Process of Omniphobic Membrane

The state of droplets on the plane can be divided into [18–20]: Young model, Wenzel model and Cassie Baxter model (Figure 8). Compared with the unmodified membrane, the omniphobic membrane meets the two conditions of high roughness and low surface energy at the same time. Attributing to the existence of an air gap, the droplets always maintain the Cassie Baxter state without contacting the membrane directly. Therefore, the membrane can achieve the property of anti-wetting even if the feed liquid contains sodium dodecyl sulfonate (SDS).

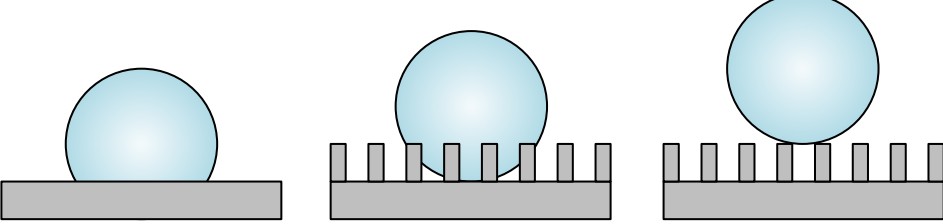

**Figure 8.** The state of droplets on the plane of the Young model, Wenzel model and Cassie-Baxter model.

After treating high salinity mine water with the PVDF-H membrane for 20 h, no obvious membrane wetting phenomenon was found. Inorganic scaling did not cause serious damage to the membrane surface structure, the COD of the feed liquid was low, and there were almost no low surface energy substances that could wet the membrane in the water. Low surface tension organics such as SDS or humic acid may be introduced in the mining process, which increases the risk of membrane wetting [21].

As shown in Figure 9a, after treating high salinity mine water containing SDS, the surface of the PVDF-H membrane was covered with a layer of white granular crystals, namely saline SDS colloidal dirt. The polar group of SDS has a negative charge. The presence of counterions in the feed will shield the electrostatic repulsion between the polar groups of SDS molecules and combine with SDS under the action of electrostatic attraction to form a gel layer which aggravates the phenomenon of wetting [22,23]. The counterions in high salinity mine water are mainly $Na^+$, $Ca^{2+}$ and $Mg^{2+}$. The binding energy and dissociation energy of $Ca^{2+}$ and $Mg^{2+}$ are higher than $Na^+$, so the membrane surface pollutants are mainly $Ca^{2+}$–$Mg^{2+}$/SDS aggregates. In contrast, only a small amount of pollutants exists on the surface of the PVDF-O membrane, indicating that the omniphobic membrane can resist surfactant wetting (Figure 9b).

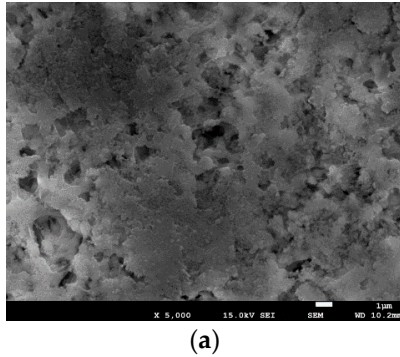
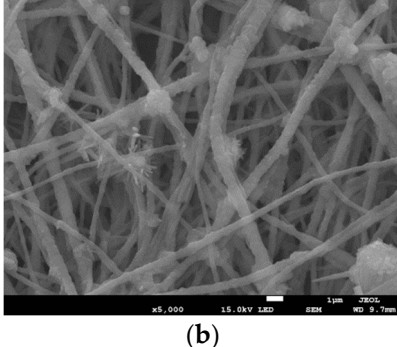

(**a**)                                                    (**b**)

**Figure 9.** Scanning electron micrograph of $Ca^{2+}$–$Mg^{2+}$/SDS fouled membranes: (**a**) PVDF-H, (**b**) PVDF-O

In addition to counterions, the concentration of the surfactant also affects the wettability of the membrane. As shown in Figure 10, according to the different positions of the gas–liquid interface, the wetting degree can be divided into the following four cases [24]: (1) not wetted, (2) surface wetted, (3) partially wetted and (4) completely wetted. The adsorption of SDS on the PVDF membrane is a concentration-dependent phenomenon. When treating high salinity mine water containing 0.1 or 0.2 mmol/L SDS, the membrane was only surface wetted, the flux decreased due to the reduction of the water gasification area, and the conductivity increased slightly; Increasing the concentration of SDS to 0.3 mmol/L, most of the pores were wetted after 140 min, and the conductivity increased significantly. Continuing to increase the concentration of SDS to 0.5 mmol/L, the conductivity increased faster (Figure 11). These results illustrated that the increase of the concentration of SDS in the feed solution will accelerate the wetting of the membrane. When the concentration was less than the critical micelle concentration (CMC), the membrane was only surface wetted.

If the concentration of SDS continues to increase, the surfactant will accumulate on the membrane surface intensively, aggravating the degree of wetting. It should be noted that both sides of the membrane are in direct contact with the hot feed liquid and the condensate during DCMD. If the membrane pores were completely wetted, water will flow from the cold side with low concentration to the hot side with high concentration according to the principle of positive permeability, resulting in negative flux.

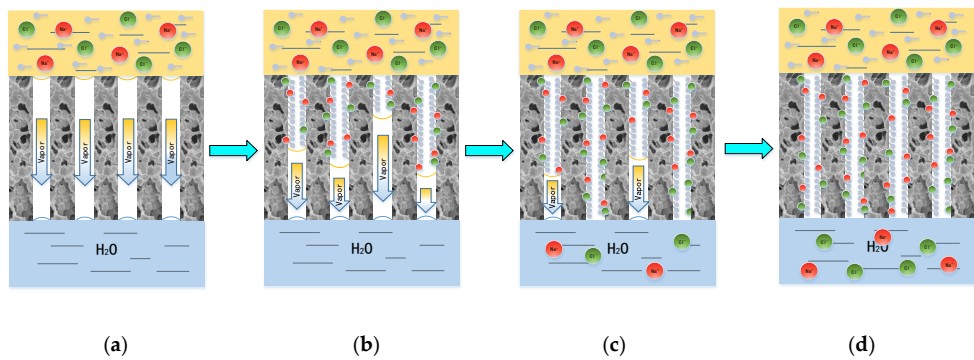

| (**a**) | (**b**) | (**c**) | (**d**) |

**Figure 10.** Surfactant wetting mechanism of the membrane: (**a**) Not wetted, (**b**) Surface wetted, (**c**) Partially wetted, (**d**) Completely wetted.

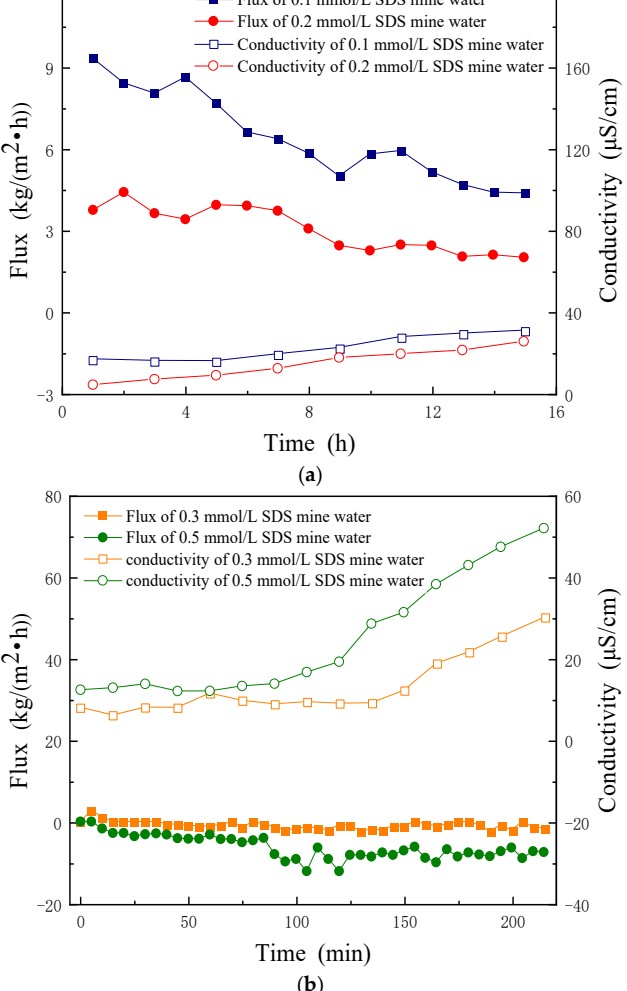

**Figure 11.** Flux and conductivity variation of PVDF-H membrane in the treatment of mine water (**a**) Containing 0.1 and 0.2 mmol/L SDS, (**b**) Containing 0.3 and 0.5 mmol/L SDS by DCMD.

When the PVDF-H membrane was used to treat high salinity mine water containing 0.1 and 0.2 mmol/L SDS, the flux decreased by 37.8% and 40% respectively within 10 h (Figure 11). When the PVDF-H membrane was used to treat high salinity mine water containing 0.3 mmol/L SDS, it will be wetted within 100 min. In contrast, the flux of mine water containing 0.1 and 0.2 mmol/L SDS treated with the PVDF-O membrane decreased by only 14% and 19.4%, the conductivity of produced water did not increase significantly within 10 h of treatment of mine water containing 0.3 mmol/L SDS, and the content of Ca element on the membrane surface decreased to 5.9%, indicating that the introduction of low surface energy substances enhanced the anti-wetting property of the membrane (Figure 12).

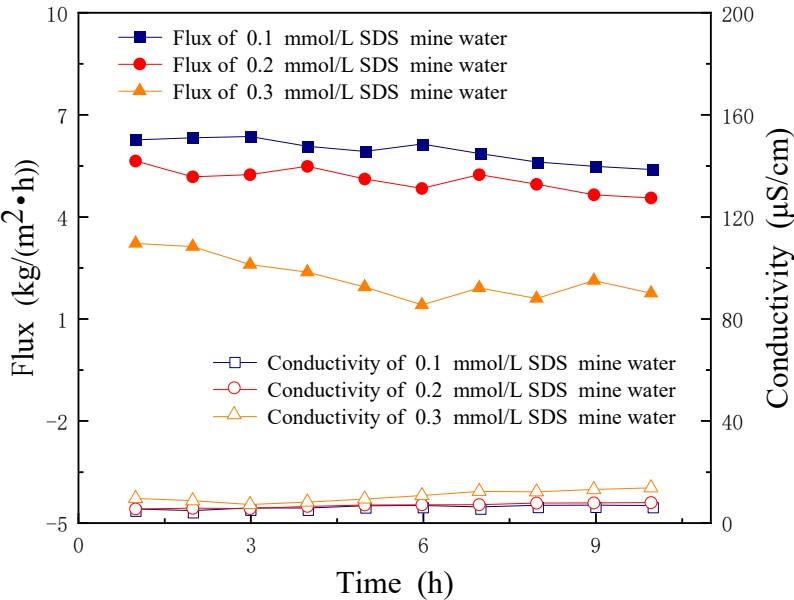

**Figure 12.** Flux and conductivity variation of PVDF-O membrane in the treatment of mine water containing 0.1, 0.2 and 0.3 mmol/L SDS by DCMD.

*3.2. Analyzing of the Interaction between Pollutants and Membrane with XDLVO Theory*

3.2.1. Effects of Different Pollutants on MD Performance

Pollutants in high salinity mine water include a large amount of $Na^+$, $Cl^-$ and calcium–magnesium salts, and a very small amount of substances such as $K^+$, $NO_3^-$ and ammoniacal nitrogen. Water with different concentrations of NaCl and different hardnesses were treated by DCMD for 20 h, as were the raw and the concentrated mine water. As shown in Figure 13, it can be observed that when treating feed containing 4 g/L or 35 g/L NaCl, the flux was similar and hardly decreased. In short-term operation, the concentration polarization caused by NaCl alone has little effect on the flux. When treating calcium–magnesium salt solution and high salinity mine water, the flux in both decreased by about 62.5%, and the scaling on the membrane surface was serious. With the increase of calcium–magnesium hardness in the feed liquid, the flux decreased faster. No obvious membrane wetting occurred after 20 h. However, some studies have also shown that the presence of inorganic scaling may accelerate the membrane wetting, and the crystalline salt can penetrate the interior of the membrane to 20–50 μm [25,26]. When the operation time is long enough, inorganic scaling will cause irreversible damage to the membrane surface and pores, the mechanical strength will gradually decrease, and the membrane may even degrade in serious cases [27]. With the deposition of the hydrophilic substance NaCl in pores, water will follow NaCl crystals into the membrane pores until it penetrates the membrane, resulting in membrane wetting.

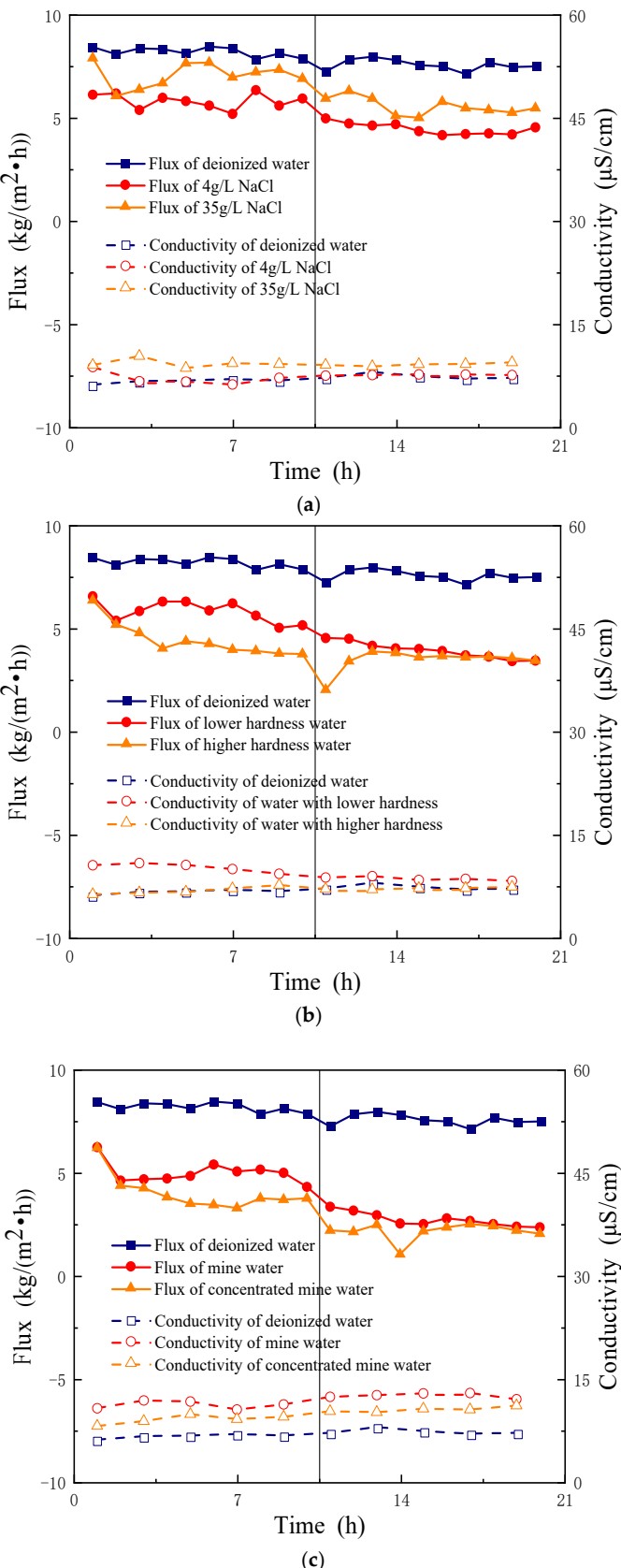

**Figure 13.** Flux and conductivity variation of the PVDF-H membrane in the treatment of (**a**) NaCl solution with different concentrations, (**b**) solution with different calcium–magnesium hardness, (**c**) mine water and concentrated mine water by DCMD.

### 3.2.2. Calculation of Interaction Energy between Different Pollutants and Membrane

According to the partial data of surface tension (Table 6), the interfacial free energy between different pollutants in water and membrane surfaces can be calculated. The calculation results are shown in Table 7. In general, a positive value indicates repulsion, a negative value indicates attraction, and the greater the absolute value, the stronger the repulsion or attraction. The total interface interaction energy between the PVDF-H membrane and pollutants is: $-46.25$, $-52.84$, $-61.12$ mJ·m$^{-2}$ respectively, indicating that there is adsorption between pollutants and the membrane surface in general; The total interfacial interaction energy of the omniphobic membrane is: $-51.80$, $-60.18$ mJ·m$^{-2}$ respectively. For the same pollutants, the absolute values of the total interfacial interaction energy of the PVDF-H membrane are higher than that of the PVDF-O membrane, indicating that the commercial membrane is more vulnerable to pollution. Due to the different surface properties of the membrane and the different types of pollutants, the values of AB, LW and EL interaction energy are also different. For either the PVDF-H or PVDF-O membrane, if LW interaction energy and EL interaction energy tend to zero, this indicates that van der Waals force and electrostatic force have little influence. The AB action energy is less than zero, resulting in adsorption, and the adsorption is far greater than the repulsion. Therefore, the Lewis acid–base action plays a leading role in the pollution process.

**Table 6.** Contact angle and surface tension of membranes and foulants.

| Samples | Contact Angles (°) | | | Surface Tension (mJ/m$^2$) | | | Zeta Potential (mV) |
|---|---|---|---|---|---|---|---|
| | $\theta_W$ | $\theta_G$ | $\theta_D$ | $\gamma_s^{LW}$ | $\gamma_s^+$ | $\gamma_s^-$ | |
| PVDF-H | 119.80 | 95.62 | 65.43 | 18.12 | 0.209 | 1.306 | $-63.9$ |
| PVDF-O | 153.48 | 111.81 | 93.79 | 12.12 | 0.541 | 1.015 | $-68.0$ |
| Foulant 1 | 71.1 | 40.6 | 37.4 | 26.78 | 2.164 | 5.205 | $-6.9$ |
| Foulant 2 | 82.8 | 46 | 41.4 | 26.74 | 2.753 | 1.717 | $-13.8$ |
| Foulant 3 | 107 | 64.5 | 57.9 | 22.55 | 3.175 | 0.080 | $-30.3$ |

Note: Foulant 1 is calcium–magnesium salt precipitation, Foulant 2 is mine water sedimentation, Foulant 3 is Ca$^{2+}$-Mg$^{2+}$/SDS.

**Table 7.** Interfacial interaction energy between the foulants and membrane surfaces. (mJ·m$^{-2}$).

| Membrane-Foulant | $\Delta G_{d0}^{LW}$ | $\Delta G_{d0}^{AB}$ | $\Delta G_{d0}^{EL}$ | $\Delta G_{d0}^{TOT}$ |
|---|---|---|---|---|
| PVDF-H-Foulant 1 | 0.417 | $-46.667$ | $-4.076 \times 10^{-4}$ | $-46.25$ |
| PVDF-H-Foulant 2 | 0.414 | $-53.256$ | $-2.213 \times 10^{-4}$ | $-52.84$ |
| PVDF-H-Foulant 3 | 0.066 | $-61.175$ | $-9.848 \times 10^{-5}$ | $-61.12$ |
| PVDF-O-Foulant 2 | 1.192 | $-52.990$ | $-2.591 \times 10^{-4}$ | $-51.80$ |
| PVDF-O-Foulant 3 | 0.189 | $-60.370$ | $-1.241 \times 10^{-4}$ | $-60.18$ |

To further understand the interaction relationship between membrane and pollutant, the interaction energy between the membrane surface and pollutant surface at different distances was calculated, and the results are shown in Figure 14. The interaction energy of AB, LW and EL approach 0 with the increase of distance. In the short-range region (when the distance between the pollutant and the membrane surface is less than 5 nm), the variation range of AB interaction energy with distance is the largest, and its curve almost coincides with the total interaction energy curve, which shows that AB interaction energy plays a leading role in the interaction between the membrane and pollutant. The interaction energy between LW and EL shows a long-range effect. In the long-range region, the curve trend is consistent with the total interaction energy curve, indicating that the repulsion or attraction effect caused by LW and EL will also affect the pollutants located in the long-range region.

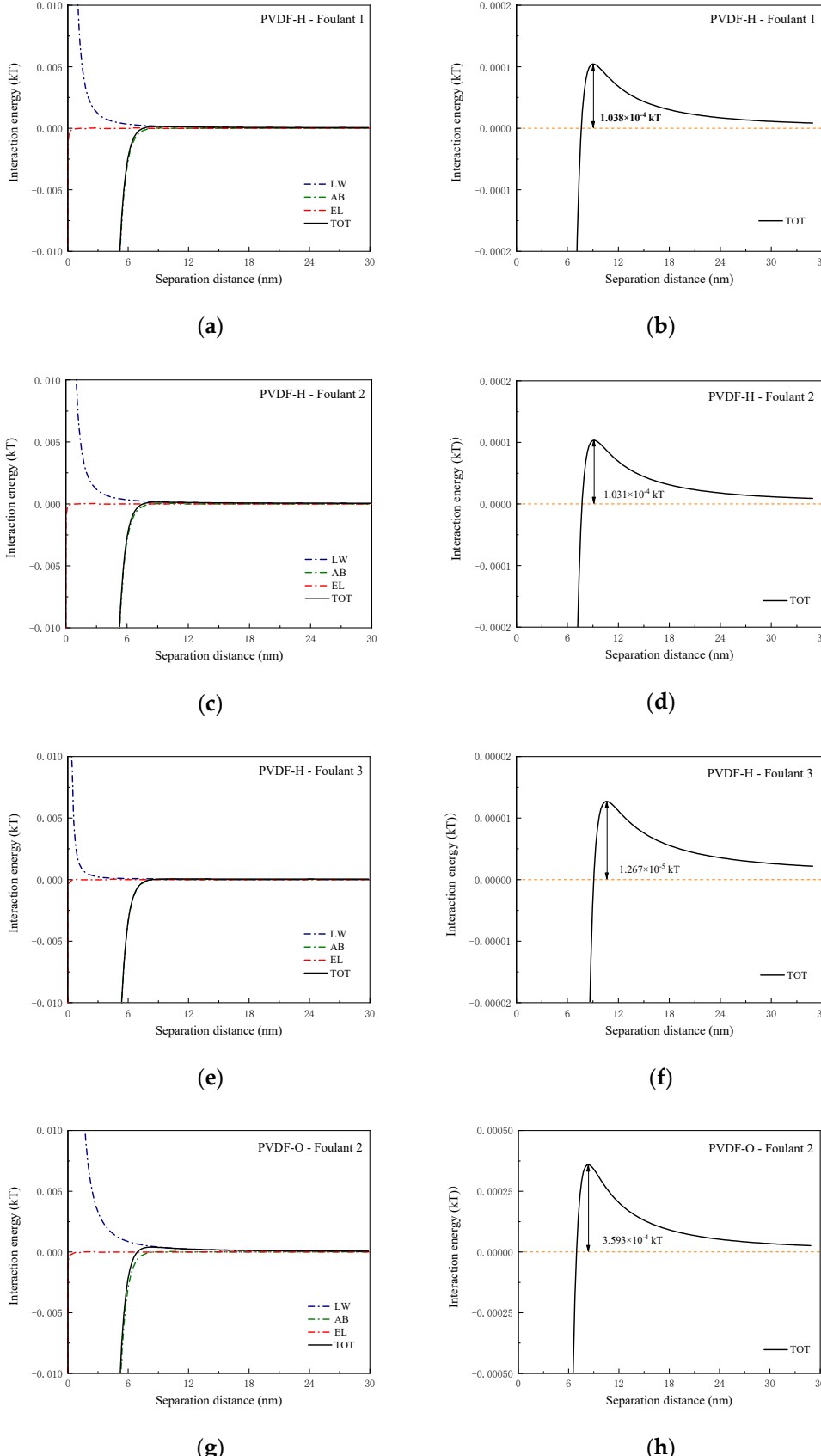

**Figure 14.** *Cont.*

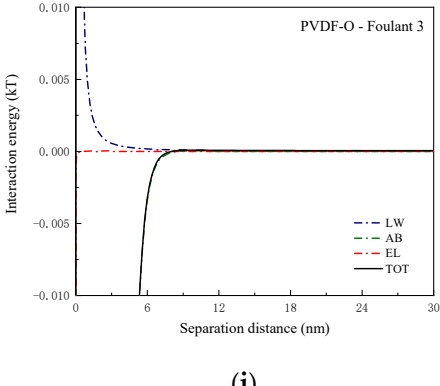 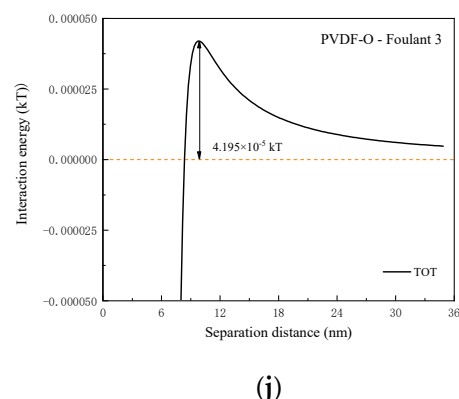

(**i**)                               (**j**)

**Figure 14.** The variation of interfacial interaction energies between pollutants and membrane surface with different spacing: (**a**,**b**) PVDF-H—Foulant 1; (**c**,**d**) PVDF-H—Foulant 2; (**e**,**f**) PVDF-H—Foulant 3; (**g**,**h**) PVDF-O—Foulant 2; (**i**,**j**) PVDF-O—Foulant 3.

In the process of pollutants approaching the membrane surface, there is a repulsive energy barrier. Pollutants can be adsorbed and deposited on the membrane surface only by overcoming this repulsive barrier. The repulsion barriers of the PVDF-H membrane are $1.038 \times 10^{-4}$ kT, $1.031 \times 10^{-4}$ kT and $1.267 \times 10^{-5}$ kT respectively. The repulsive barrier between $Ca^{2+}$–$Mg^{2+}$/SDS and the membrane is the smallest, so it is easier for $Ca^{2+}$–$Mg^{2+}$/SDS to be adsorbed on the membrane surface, resulting in membrane fouling and wetting. The repulsion barriers of the PVDF-O membrane are $3.593 \times 10^{-4}$ kT and $4.195 \times 10^{-5}$ kT respectively. The repulsion barrier between pollutants and the PVDF-O membrane is higher than it is for the PVDF-H membrane, so it is difficult for pollutants to be adsorbed on the omniphobic membrane. According to XDLVO, the conclusion is consistent with the changing trend of flux in the process of DCMD, which means it is convincing to be utilized to analyze membrane fouling and wetting.

## 4. Conclusions

(1) The PVDF omniphobic membrane achieved the properties of anti-wetting and anti-fouling after modification, improving the performance of MD in treating high salinity mine water. The XDLVO theory provided more powerful proofs that the existence of $Ca^{2+}$–$Mg^{2+}$/SDS aggravated the risk of membrane-wetting and the omniphobic membrane cannot be fouled or wetted easily due to its higher repulsion barrier between pollutants and the membrane. It has a guiding role for the advanced treatment of high salinity mine water. (2) The omniphobic membrane can be used not only for the treatment of mine water, but can also be applied to the treatment of other kind of wastewater containing low surface energy pollutants. In the process of the modification, if the micro-nano-concave structure was constructed on the membrane surface before fluorination, it is expected to realize the wetting resistance to higher concentrations of organic pollutants. If the pore size and pore size distribution were measured, it may better reflect the performance index of the membrane.

**Author Contributions:** Y.W.: Conceptualization, Methodology, Mathematical calculation, Writing-Original Draft. Z.L. and K.X.: Validation, Formal analysis. R.H.: Data Curation. Q.G. and J.L.: Supervision. S.L.: Project administration. All authors have read and agreed to the published version of the manuscript.

**Funding:** This research was supported by Science and technology innovation support project of national energy group (GJNY2030XDXM-19-04.1) and China University of Mining and Technology (Beijing) Yueqi Outstanding Scholar Project (Project Number 2020JCB02).

**Acknowledgments:** The authors thank The Open Fund Project of the State Key Laboratory of Water Resources Protection and Utilization in Coal Mining for funding this project.

**Conflicts of Interest:** The authors state that they have no known competing financial interests or individual relationships that could have appeared to affect the work reported in this article.

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
