# Peer review of "Preparation and Modification of PVDF Membrane and Study on Its Anti-Fouling and Anti-Wetting Properties"

_water, doi:10.3390/w14111704_

Round 1

Reviewer 1 Report

The paper discuss the preparation and modification of PVDF membrane with the focus on anti-fouling and anti-wetting properties. It presents compact state-of-the-art and clear preparation and testing procedure. It is written in good English. However, the discussion of results is not well presented - data are described with not enough care and precision - this section needs to be enhanced. 

Detailed comments are presented below:

L17: membrane was not prepared in this paper, the method of preparation was presented in this paper

L40: direct

L92: What is the purpose of using sonication in membrane preparation?

Table 1:  The presented components were prepared separately, measured and then mixed? Was each of the component dissolved in 1L of water or is it the total amount of a given component in 1L of feed? Please give more details.

L137-142: The explanation of symbols used should be placed right after the first mention. Thus this description should be moved to the section starting with line 131.

L147: What is K in this formula? Is does not appear in any other equation.

L151: Authors used zi in equations and Zi in description. Please choose one.

L216: hydroxyl groups?

L222: Fouling is a process of clogging of MEMBRANE not an inorganic salt.

L250: Young model

Fig.11 (b) : How can the measurements be interpreted if according to the chart the flux is 0 or even negative??? There is not even a word of comment to this situation…

Fig.14: This figure contains a lot of charts and the caption is very enigmatic. For more clarity Authors should give more detailed caption: as the charts are described as made for different spacing, at least information about this spacing should be given.

Conclusion section: I believe that Authors can give more extensive comment to presented results.

Reviewer 2 Report

The following article was submitted for review: Preparation and modification of PVDF membrane and study on its anti-fouling and anti-wetting properties. This is a very interesting issue. The test results can be valuable and used in cleaning processes.

  1. However, the abstract of the article does not show the essence of the problem. The aim and conclusions of the study must be clearly visible. This version is chaotic. There are no explained abbreviations which makes it difficult to review. We find them only in the introduction, but not all of them.
  2. Tests and test results are presented correctly. However, please pay attention to the captions under the figures. From Figure 6, a different signature format was used than for the previous figures. The same is true for tables. From table 4 we have a different format for signing tables. Please also correct the caption under Figure 7.
  3. Please correct the conductivity unit notation in the table 1and the pH.
  4. In my opinion, the publication is clearly written, the list of experiments and the analysis of the results are correct. Detailed conclusions were drawn. The literature is properly selected.
  5. After considering the comments, the work will be suitable for publication.
  6. Apart from the minor mistakes I mentioned earlier, the publication is prepared very carefully.

Thank you for considering my opinion. I encourage authors to keep on working to improve the manuscript.

Reviewer 3 Report

The manuscript on the PVDF membranes applied for the membrane distillation of various types of polluted waters, is more appropriate for Membranes journal. Unfortunately, the research is described in an unclear way which makes the understanding of the manuscript quite difficult (if not impossible).
At the submitted version I would not recommend the manuscript for the publication. The following questions/comments could be addressed to the Authors.
1. Abstract is unclear, reader can get an impression that only 1 membrane was prepared, but Authors described 3 various membranes (PVDF-H, PVDF-E, and PVDF-O).
2. There is no explanation of the abbreviation of XDLVO theory (neither in Abstract nor in the text). In fact, I am not sure if the part devoted to the application of XDLVO theory is needed.
3. Lines 46-66 - information regarding MD are incorrect. MD is NOT a new process (it has been known for more than 50 years already). The driving force is NOT vapor pressure deficit. There are NO "holes" in the membrane.
4. Section 2 - no clear information regarding the commercial membrane used. What is 17-FAS? What type of the equipment was used for the electrospinning? Table 1 - it should be "pH" not "PH". The concentration of solutions should be written as "wt%" not "ωt%" (e.g., line 88, 92, Fig. 1....).
5. Results and discussion - Tab. 3 - wrong header. What was the accuracy of the WCA measurements? Authors should not present results with 2 decimal places. Usually the error is in the range of 2-3°.
6. There is no information regarding the pore size and pore size distribution for all investigated membranes.
7. Conclusions shouldn't repeat the information presented in Abstract.
8. References - 5 out of 30 are published in Chinese making them useless for other readers. The journal is an English language journal and all cited papers must be also in English. Moreover, there are much more important papers related to the subject written by authors from Europe or Americas. It must be also taken into account.

Reviewer 4 Report

The paper is interesting.
I recommend publication only if the following issues can be addressed.

- Lines 40-41 page 1: You should mention that discharge of mine water (brine) degrades water quality and thus water cannot be directly used for potable water (via desalination) and industrial applications. Cite the following references:

Panagopoulos, A. (2021). Techno-economic assessment of Minimal Liquid Discharge (MLD) treatment systems for saline wastewater (brine) management and treatment. Process Safety and Environmental Protection, 146, pp. 656-669. 

Panagopoulos, A. (2022). Techno-economic assessment of zero liquid discharge (ZLD) systems for sustainable treatment, minimization and valorization of seawater brine. Journal of Environmental Management, 306, 114488.

Panagopoulos, A. (2022). Brine management (saline water & wastewater effluents): Sustainable utilization and resource recovery strategy through Minimal and Zero Liquid Discharge (MLD & ZLD) desalination systems. Chemical Engineering and Processing - Process Intensification, 108944.

- Much more explanations and interpretations must be added for the Results

- How many replications you performed for your experiments?

- The authors should provide error bars for all the data

- Conclusion: Discuss the applicability of your findings/results and future study in this field.

- Conclusion: Make it as one or two paragraphs.

- Language editing is recommended.

- What is the ion composition of the feed water?

Round 2

Reviewer 1 Report

I am satisfied with Authors' answers. 

Author Response

Thank you very much !

Reviewer 3 Report

Authors tried to improve the manuscript, unfortunately the part related to membrane distillation was not improved. As I stated in my previous review MD is NOT driven by "vapor pressure deficit" but by the difference in the chemical potential of volatile components on the both sides of the membrane. Moreover, discussing membranes, we cannot use term "hole(s)" - in membrane science we have nonporous membranes or membranes possessing pores (of different size). I did not get an answer for the comment on the type and producer of the electrospinning equipment. In the text Authors added information about the commercial PVDF membrane, but it is also called "unmodified", similarly as PVDF-E. It is suggested to remove the word "unmodified" when talking about PVDF-H (commercial) membrane.

Reviewer 4 Report

The authors improved significantly their manuscript, which is now ready for publication. I recommend acceptance for publication.

Author Response

Thank you very much !